

# Net land-use change carbon flux estimates and sensitivities - An assessment with a bookkeeping model based on CMIP6 forcing

Kerstin Hartung[1,*], Ana Bastos[1,2], Louise Chini[3], Raphael Ganzenmüller[1], Felix Havermann[1], George C. Hurtt[3], Tammas Loughran[1], Julia E.M.S. Nabel[4], Tobias Nützel[1], Wolfgang A. Obermeier[1], and Julia Pongratz[1]

[1]Ludwig-Maximilians-Universität München, Munich, Germany

[2]Max Planck Institute for Biogeochemistry, Jena, Germany

[3]Department of Geographical Sciences, University of Maryland, U.S.A.

[4]Max Planck Institute for Meteorology, Hamburg, Germany

[*]Now at: Deutsches Zentrum für Luft- und Raumfahrt, Institut für Physik der Atmosphäre, Oberpfaffenhofen, Germany

**Correspondence:** Kerstin Hartung (kerstin.hartung@dlr.de)

**Abstract.** The carbon flux due to land-use and land-cover change (net LULCC flux) historically contributed to a large fraction of anthropogenic carbon emissions while at the same time being associated with large uncertainties. This study aims to compare the contribution of several sensitivities underlying the net LULCC flux by assessing their relative importance in a bookkeeping model (BLUE) based on a LULCC dataset including uncertainty estimates (the LUH2 dataset). The sensitivity experiments

build upon the approach of Hurtt et al. (2011) and compare the impacts of LULCC uncertainty (a high, baseline and low land-use estimate), the starting time of the bookkeeping model simulation (850, 1700 and 1850), net area transitions versus gross area transitions (shifting cultivation) and neglecting wood harvest on estimates of the net LULCC flux. Additional factorial experiments isolate the impact of uncertainty from initial conditions and transitions on the net LULCC flux. Finally, historical simulations are extended with future land-use scenarios to assess the impact of past LULCC uncertainty in future projections.

Over the period 1850–2014, baseline and low LULCC scenarios produce a comparable cumulative net LULCC flux while the high LULCC estimate initially produces a larger net LULCC flux which decreases towards the end of the period and even becomes smaller than in the baseline estimate. LULCC uncertainty leads to slightly higher sensitivity in the cumulative net LULCC flux (up to 22 %, reference are the baseline simulations) compared to the starting year of a model simulation (up to 15 %). The contribution from neglecting wood harvest activities (up to 28 % cumulative net LULCC flux) is larger than from

LULCC uncertainty and the implementation of land-cover transitions (gross or net transitions) exhibits the smallest sensitivity (up to 13 %). At the end of the historical LULCC dataset in 2014, the LULCC uncertainty retains some impact on the net LULCC flux ($\pm 0.15$ PgC yr$^{-1}$ at an estimate of 1.7 PgC yr$^{-1}$). Of the past uncertainties in LULCC, a small impact persists in





2099, mainly due to uncertainty of harvest remaining in 2014. However, compared to the uncertainty range of the LULCC flux estimated today, the estimates in 2099 appear to be indistinguishable.

These results, albeit from a single model, are important for CMIP6 as they compare the relative importance of starting year, uncertainty of LULCC, applying gross transitions and wood harvest on the net LULCC flux. For the cumulative net LULCC flux over the industrial period the uncertainty of LULCC is as relevant as applying wood harvest and gross transitions. However, LULCC uncertainty matters less (by about a factor three) than the other two factors for the net LULCC flux in 2014 and historical LULCC uncertainty is negligible for estimates of future scenarios.

## 1 Introduction

Globally, the historical net carbon flux due to land-use and land-cover change (net LULCC flux) has been positive (i.e. a source: flux from land to the atmosphere). The net LULCC flux makes up a considerable part of overall anthropogenic carbon emissions and is associated with large uncertainties. During the period 2009–2018, Friedlingstein et al. (2019) estimate the

net LULCC flux to be $1.5 \pm 0.7$ PgC yr$^{-1}$ and to make up about 14% of total anthropogenic carbon emissions. Alternatively, Lawrence et al. (2016) discuss different estimates of the contribution from the net LULCC flux to total anthropogenic emissions of up to 45 %, depending on the details of the comparison (years and flux components considered).

  Since the net flux from LULCC cannot be directly measured, we can only rely on values calculated by models, for example dynamic global vegetation models (DGVMs) and bookkeeping models. Bookeeping models (Houghton, 2003; Houghton and

Nassikas, 2017; Hansis et al., 2015) combine observation-based carbon densities with LULCC estimates to determine the net LULCC flux. DGVMs, on the other hand, model the evolution of carbon pools on a process-based level and also react to climate impacts and trends.

  Differences in model estimates of the net LULCC flux can have different origins, broadly falling into three categories: (i) the underlying LULCC reconstruction and its uncertainties, (ii) the LULCC practises considered (e.g. wood harvest and

shifting cultivation) and (iii) model assumptions (e.g. parameterizations of processes like type and lifetime of wood products). Considering point (i), several global multi-century LULCC reconstructions exist (i.e. Pongratz et al., 2008; Kaplan et al., 2011; Klein Goldewijk et al., 2017; Hurtt et al., 2020). Furthermore, several studies isolated and quantified the impact on the net LULCC flux of the individual components of the three categories listed above: (i) the impact of the choice of LULCC dataset (Hurtt et al., 2006; Pongratz et al., 2008; Stocker et al., 2011); (ii) the importance of neglecting or modelling wood

harvest (Stocker et al., 2014; Arneth et al., 2017) and shifting cultivation (Hurtt et al., 2011; Wilkenskjeld et al., 2014; Stocker et al., 2014; Arneth et al., 2017); and (iii) the model assumptions, for example using either DGVMs or bookkeeping models (Houghton et al., 2012; Gasser et al., 2020). The starting year of a simulation can either be seen as part of the LULCC itself (category i) or a model assumption (category iii).



Example studies comparing the impact of properties across at least two of these categories are Hurtt et al. (2011) and Gasser
et al. (2020). In the Hurtt et al. (2011) sensitivity study based on the LUH1 dataset (Land-Use Harmonization, Chini et al.,
2014), the authors analysed over 1600 simulations with respect to model "factors" like the simulation start date, the choice
of historical and future agricultural land-use and wood harvest scenarios, and inclusion of shifting cultivation. The simulation
outputs were compared across a variety of metrics and diagnostic tools including secondary area and mean age, global gross and
net transitions, and cumulative gross and net loss of aboveground biomass. Their analysis showed that the most relevant factors
were the start date, and the inclusion of both shifting cultivation and wood harvesting. The LUH2 dataset (Hurtt et al., 2020)
responded to these findings by developing a dataset that started in 850, with improved representations of the spatial patterns
of both shifting cultivation and wood harvesting based on remote-sensing data. Gasser et al. (2020) use a hybrid model (the
OSCAR model) combining bookkeeping properties (tracking the effect of LULCC activities) and biogeophysical properties
from a DGVM to estimate uncertainties acting on annual and cumulative $CO_2$ emissions. The focus in Gasser et al. (2020) is
on the relative importance of biogeophysical parameters, the LULCC dataset (either the LUH2 or the FRA (Forest Resources
Assessment, fao, 2015) dataset) and the inclusion of the LASC (loss of additional sink capacity, Pongratz et al., 2014 ) to the
net LULCC flux. The latter property constitutes one of the main differences of the resulting flux estimates between DGVMs
and bookkeeping models and is due to changes in carbon densities caused by varying atmospheric $CO_2$ concentrations. Gasser
et al. (2020) find that the largest variation in flux estimates is induced by biogeophysical parameters (mainly carbon densities),
followed by the definition of the LULCC flux (i.e. including or excluding LASC). The LULCC dataset is found to cause the
least uncertainty cumulatively, though the trend of the annual LULCC flux based on the two datasets has opposing signs in
recent years.

The goal of our study is to build upon previous approaches (e.g. Hurtt et al., 2011 and Gasser et al., 2020) to assess a variety
of the above mentioned sensitivities of the net LULCC flux with one single underlying LULCC dataset reporting uncertainty
(LUH2, Hurtt et al. 2020) and the bookkeeping model BLUE (bookkeeping of land use emissions, Hansis et al. 2015). The
LUH2 dataset (Hurtt et al., 2020) provides historical land-use estimates from 850 with uncertainty estimates for agricultural
land area (from the History Database of the Global Environment (HYDE), Klein Goldewijk et al. 2017) and wood harvest (Zon
and Sparhawk, 1923; Kaplan et al., 2017). The dataset captures the challenge of reconstructing the LULCC of the past. LUH2 is
the land-use dataset that is applied in CMIP6 (Eyring et al., 2016) for simulations with process-based DGVMs, like in LUMIP
(Land Use Model Intercomparison Project, Lawrence et al. 2016). BLUE is a data-driven bookkeeping model (Hansis et al.,
2015) used in the annual global carbon budget (GCB) for LULCC flux estimates (Friedlingstein et al., 2019). We choose a
bookkeeping model in contrast to a DGVM because LULCC fluxes due to individual LULCC events can be traced and because
of the potential to isolate the net LULCC flux independent of climate variability, among other factors (Pongratz et al., 2014).

Due to the high computational efficiency of the bookkeeping model, several sensitivity experiments can be produced and an
exhaustive comparison of common factors impacting the total net LULCC flux is possible. Here, the impact of modelling wood
harvest and shifting cultivation as land management processes is compared to the impact of uncertainties of the LULCC dataset
and the initialisation year of the LULCC simulation. We design additional artificial sensitivity experiments to disentangle the
uncertainty from the initial land cover distribution and the uncertainty from LULCC activities (transitions). By extending the





historical simulations under future LULCC scenarios, we can then estimate the impact of past uncertainty on future estimates
of the net LULCC flux. Our study thus provides an extension to previous studies comparing sensitivities across a different set
of factors by also disentangling the relevance of the initial land-cover distribution compared to the uncertainties in LULCC
activities. In addition, it updates the sensitivities of e.g. wood harvest and shifting cultivation based on a more recent LULCC
dataset, which is also the basis for CMIP6.

The analysis of the simulations is guided by two main questions: 1. How do LULCC uncertainties influence the overall
emitted carbon? and 2. What uncertainties remain at the end of the historical period and how much do they influence future
projections? For both questions the global net LULCC flux, as well as separation by LULCC activity and by different regions
are considered. This analysis can serve as reference for subsequent sensitivity analyses with complex models (DGVMs, ESMs)
and points to model and data choices which matter most for modelling of land-use related changes in the carbon cycle.

## 2   Model description, LULCC dataset and experiment setup

As a first step, we present the bookkeeping model BLUE used in this study. Then the LUH2 dataset, its high and low LULCC
scenarios as well as various future scenarios are introduced. Finally, an overview of the conducted BLUE experiments is given.
A brief description of how the LUH2 dataset is prepared for use with the BLUE model and short discussion of the properties
of the LULCC dataset are provided in the Appendix (Sec. A1 and A2).

### 2.1   The bookkeeping model BLUE

BLUE (Hansis et al., 2015) is a data-driven, semi-empirical bookkeeping model. Initial areas of the four land-cover types
primary land, secondary land, cropland and pasture determine the amount of carbon stored in soil and vegetation biomass
prior to tracked LULCC activities. These initial "equilibrium pools" are determined from observation-based carbon densities
and are non-zero for the carbon associated with the soil component undergoing slow relaxation processes and the vegetation
biomass. LULCC activities, i.e. land-use transitions, take the model state away from equilibrium, increasing or decreasing
so-called disequilibrium pools. BLUE considers the four LULCC activities abandonment (cover change from crop or pasture
to secondary land), clearing for cropland or pasture (cover change from primary land, secondary land, crop or pasture to crop
or pasture) and wood harvest (cover change from primary to secondary land, or land management on secondary land). As wood
harvest is the only type of harvest modelled in BLUE it is in the following abbreviated as harvest. Disequilibrium pools exist
for vegetation, soil undergoing fast and slow relaxation processes, and for products from harvest and clearing with lifecycles
of 1, 10 and 100 years. Response curves characterise the temporal adjustment of the disequilibrium pools after a transition to
the new equilibrium, where the difference in carbon stocks, namely the content of the disequilibrium pools, is steadily emitted
to the atmosphere. The version of BLUE used here and in the GCB is based on 11 natural plant functional types (PFTs), of
which six represent forested biomes, and two agricultural PFTs (crop and pasture). More information on the BLUE model can
be found in Hansis et al. (2015).





For the analysis it is useful to note a few additional model assumptions. If two simulations are based on the same LULCC dataset but start in different years ($y_2 > y_1$), then areas of the four cover types will be identical in year $y_2$ but the disequilibrium carbon pools and the resulting flux to the atmosphere will not be identical. As the simulation started in $y_2$ is based on the initial land cover of that year, it will only track LULCC activities occurring after $y_2$ and not all the activities that have happened since $y_1$, as in the first case. Moreover, two simulations can have an identical cumulative LULCC flux up to a given year, but because

they might be associated with different disequilibrium pools, the subsequent evolution of fluxes can differ (see for example Fig. 2). This also applies to net LULCC flux caused by LULCC activities during the simulation but occurring after the end of the simulation (e.g. decay of long-lived harvested wood products), which are not tracked in the applied setup of the BLUE model. The first assumption, to only track LULCC activities subsequent to the start year, is specific to the model world. The second assumption, to only account for the net LULCC flux which already happened and not for the total net LULCC flux that

a LULCC activity causes, is more common also to policies.

## 2.2   The LUH2 dataset

Detailed descriptions of the LUH2 data, the agricultural area dataset HYDE and their uncertainty assessments are given in Klein Goldewijk et al. (2017) and Hurtt et al. (2020) for HYDE3.2 and LUH2, respectively.

    The LUH2 historical dataset provides a timeseries of annual, fractional land use and gross transitions on a $0.25° \times 0.25°$ grid

for the period 850–2015, though transitions are only available until 2014. Land use is characterised as urban land, cropland (annual and perennial C3 and C4 crops as well as C3 nitrogen fixers), managed pasture and rangeland, as well as natural vegetation (forested and non-forested, primary and secondary land). Rangelands are distinguished from managed pastures by an aridity index and population density from the HYDE dataset and can imply a land-cover change (e.g., in Brazil's Cerrado), but can also simply mean a different management of the original land-cover type (e.g. in the semi-dry regions of Australia). In

addition to five wood harvest transitions on primary and secondary, forested and non-forested land (for secondary forested land is further divided by forest age), gross land-use transitions are available between the different land-use types. Wood harvest is characterised alternatively by the harvested area or the removed biomass. Land-use states and transitions are available for a baseline scenario and two additional scenarios which in this study are used to quantify the uncertainty of the LULCC dataset: a high scenario assumes more land-use activity at the start of the LULCC dataset in 850 than in the baseline, whereas the low

scenario starts off with less land-use activity, and vice versa at the end of the dataset.

    The uncertainty in agricultural area is estimated in the HYDE dataset and linked to population uncertainty. The latest version of the HYDE dataset, HYDE3.2, provides data every 100 years until 1700, every 10 years between 1700 and 2000, and every year after 2000. The LUH2 dataset uses agricultural data from the uncertainty range *A* of the HYDE product, an uncertainty range based on literature and expert judgement. The uncertainty in primary/secondary land is estimated in the LUH2 dataset,

partly through application of three different wood harvest estimates based on two different datasets before 1920 (Zon and Sparhawk, 1923; Kaplan et al., 2017) and partly through the different gross transitions arising from the different LULCC time series. For the LUH2 dataset, HYDE data is interpolated and combined with annual wood harvest data from Food and Agriculture Organization (FAO) to provide annual states and transitions.





Results from four future scenarios are also included in this analysis, namely two SSP4 scenarios, SSP4-3.4 and SSP4-6.0, by
GCAM and two SSP5 scenarios, SSP5-8.5 and SSP5-3.4OS, by MAgPIE (Riahi et al., 2017; Popp et al., 2017; Calvin et al.,
2017; Hurtt et al., 2020) for the period 2015–2100. SSP4 describes an inequality scenario with low challenges to mitigation
and high challenges to adaptation. SSP5, on the other hand, is characterised by fossil-fuelled development with high challenges
to mitigation and low challenges to adaptation. In the following, the scenarios are referred to by their Shared Socioeconomic
Pathways (SSPs) and their Representative Concentration Pathways (RCPs) and not mainly by the Integrated Assessment Model
(IAM) that produced them, i.e. GCAM or MAgPIE. Hurtt et al. (2020) gives a more detailed summary of the properties of the
different land-use scenarios. Of all available future scenarios these four were selected for this study because they are based
on the same two SSP scenarios but describe a range of possible RCP scenarios. For each of the different scenarios no further
uncertainty ranges are provided but the set of scenarios is used to explore the impact of past LULCC uncertainties on the future
net LULCC flux. More information is given in Appendix A2.

It should be noted that the LUH2 dataset, as proposed by CMIP6, does not capture the full range of uncertainty but is
an estimate based on the available data (Klein Goldewijk et al., 2017; Hurtt et al., 2020). Importantly, annual updates to the
LUH2 data, for use in the GCB, are provided when further/new information becomes available, and customized versions of the
LUH2 data have been produced for use in specific studies (e.g. Frieler et al., 2017). In particular, the last years of the baseline
LUH2 scenario have been substantially revised for subsequent analyses related to the annual GCB. This includes updates in
the underlying agricultural data from the FAO, but also revisions of regionally inconsistent data (e.g. erroneous data in Brazil
in the GCB 2018 results, Le Quéré et al., 2018; Bastos et al., 2020). These corrections are not included in the current CMIP6
dataset.

## 2.3 Experimental setup and analysis

**Table 1.** Naming of the main experiments based on LUH2 scenarios with low, baseline and high LULCC and three different starting years.

| LULCC | 850 | 1700 | 1850 |
|---|---|---|---|
| low | LO850 | LO1700 | LO1850 |
| baseline | REG850 | REG1700 | REG1850 |
| high | HI850 | HI1700 | HI1850 |

We conduct 39 historical (from 850, 1700 or 1850 until 2015) and 12 future (2015–2100) simulations to quantify the
relative importance of the uncertainty in the LULCC dataset on the historical net LULCC flux with respect to other common
uncertainties. Although land-use states are available until 2015 and 2100, the net LULCC flux based on the temporal change
in carbon pools can only be calculated until 2014 and 2099, respectively. In all experiments, the model is run at the spatial
resolution of the LUH2 dataset ($0.25° \times 0.25°$) with an annual timestep.
The nine main experiments (Table 1) combine the uncertainty of the LUH2 scenarios (REG, LO and HI scenarios) with
different starting years and thus allow to compare the relative uncertainty due to LULCC with the starting year (StYr). StYr



is varied between two pre-industrial years (850 and 1700) and one marking approximately the beginning of the industrial era (1850). While most CMIP6 historical model simulations start in 1850, previous studies discuss potential problems of initialising the model in 1850. For example, Pongratz et al. (2009) found evidence for substantial anthropogenic emissions already before 1850, which is commonly associated with pre-industrial conditions. Another example are DGVM simulations conducted for

the annual GCB: since GCB2018 (Le Quéré et al., 2018) simulations are started in 1700 in order to reduce model initialisation effects in the time-spans considered (i.e. after 1850). The three starting years chosen here thus represent a range of options from the literature. All nine main simulations are produced by taking into account wood harvest and gross transitions as provided by the LUH2 dataset. REG1700 corresponds to the scenarios used in the GCB and is considered the standard experiment.

It should be noted that even for the nine main experiments differences between the area evolution in BLUE compared to the

LUH2 dataset occur. These are mainly due to mismatches in PFT between the LUH2 (harvest) input and the BLUE model. For simulations started in 1700 and 1850 the difference in primary land extent in 2014 is at most 4 % (Fig. A5), which is also true for REG and LO in 850. However, HI850 does end with about 12 % more primary land in 2014 than the LUH2 dataset. In all cases the amount of primary land is larger in BLUE than in the original LUH2 dataset, at the cost of other land-cover types. Overall this means that the total amount of net LULCC flux will be underestimated in BLUE, the most in the HI850

experiment. More information is provided in Section A2.

**Table 2.** Overview of additional sensitivity experiments. If not specified otherwise, simulations are conducted with all three starting years 850, 1700 and 1850, and simulated for HI, REG and LO. The two setups with changes to initial conditions (IC) and transitions (Trans) modify the LUH2 dataset and are artificial. Future scenarios are continued from simulations with starting year 1700 for all three LULCC experiments.

| Name | Description of sensitivity experiments |
|---|---|
| net | Net transitions (only 1700). |
| NoH | No wood harvest. |
| IC | Initial conditions from HI or LO and transitions from REG. |
| Trans | Initial conditions from REG and transitions from HI or LO. |
| SSP4-3.4 | Scenario 2014-2099 based on GCAM with RCP3.4. |
| SSP4-6.0 | Scenario 2014-2099 based on GCAM with RCP6.0 (baseline). |
| SSP5-3.4OS | Overshoot scenario 2014-2099 based on MAgPIE with RCP3.4. |
| SSP5-8.5 | Scenario 2014-2099 based on MAgPIE with RCP8.6 (baseline). |

In addition to the nine main experiments, we conduct 30 sensitivity experiments (Table 2) in order to (i) compare the sensitivity due to LULCC and StYr to other LULCC properties and (ii) to assess how historical uncertainty propagates into future scenarios.

By neglecting information on some of the LULCC activities from the input dataset, simulations without wood harvest and

with net instead of gross transitions can be produced (see Table 2). Note that the net LULCC flux is an aggregate of all sources





and sinks due to LULCC in one year and is not linked to net transitions, i.e. net and gross land-use transitions must not be mixed up with the net or gross LULCC flux.

The three LUH2 LULCC estimates differ not only in the temporal evolution of the LULCC activities but also in their initial areas, especially when the simulation starts after 850. To disentangle these effects, we conduct additional BLUE simulations based on artificial LULCC information which is not proposed by LUH2. Instead, it uses the original (REG) area initial conditions and adapted transitions (HI or LO) in experiments called Trans, or vice versa for initial conditions (IC) sensitivity experiments. Thus, the areas of these simulations in 2014 are not consistent with the LUH2 dataset (see Fig. A4). Indeed, the IC experiment with LO initial conditions and the Trans experiment with HI amount of LULCC activities deviate significantly from the range of agricultural area in the main experiments HI, REG and LO. This difference between simulations is associated with increased remaining primary land areas. Compared to REG1700, the difference in primary land in 2014 is about 10-20 %, with lower values for HI IC and LO Trans setup initialised in 1700. Smallest differences in IC and Trans experiments are found for the deviations from REG850 and differences in REG1850 are of similar magnitudes as discussed for REG1700. However, these deviations of primary land area from the LUH2 dataset are still smaller than those caused by neglecting wood harvest (not shown).

Each of the three main simulations with starting year 1700 (Table 1) is continued following each of the four future land-use scenarios until 2100 (Table 2) to produce a total of twelve simulations for the period 2015–2100 (with net LULCC flux calculated for the period 2015–2099). The experiments continuing from HI1700 and LO1700 are artificial because the area distribution of land-cover types is not set up to match in 2014/2015. However, differences in agricultural land are small (Fig. A1) and changes in forest transitions are larger in future scenarios than differences between LULCC scenarios in 2014 (not shown).

## 3   Results and discussion

The timeseries of all three historical uncertainty estimates (Fig. 1) shows the known feature of a peak in 1960 (Hansis et al., 2015; Friedlingstein et al., 2019). Before around 1960, the net LULCC flux is almost continuously rising and levels decrease after 1960 to the end of the historical LULCC dataset in 2014. Around 2000 the annual net LULCC flux is of similar magnitude as in the early 20$^{\text{th}}$ century.

### 3.1   How do LULCC uncertainties influence overall emitted carbon?

#### 3.1.1   Temporal variability of uncertainty

The temporal evolution of the cumulative net LULCC flux in the nine main simulations (Fig. 2) exhibits three central features over the common period 1850–2014: (1) the cumulative net LULCC flux is similar in experiments REG and LO. (2) Starting a simulation in 1850, rather than earlier, leads to a larger cumulative net LULCC flux over the period 1850–2014. Finally, (3) although HI produces the largest net LULCC flux initially, this is not true throughout and especially at the end of the simulation.



The increased land-use dynamics in LO in later times let LO exceed HI in terms of cumulative net LULCC flux at some point in time, which we will call a crossing point.

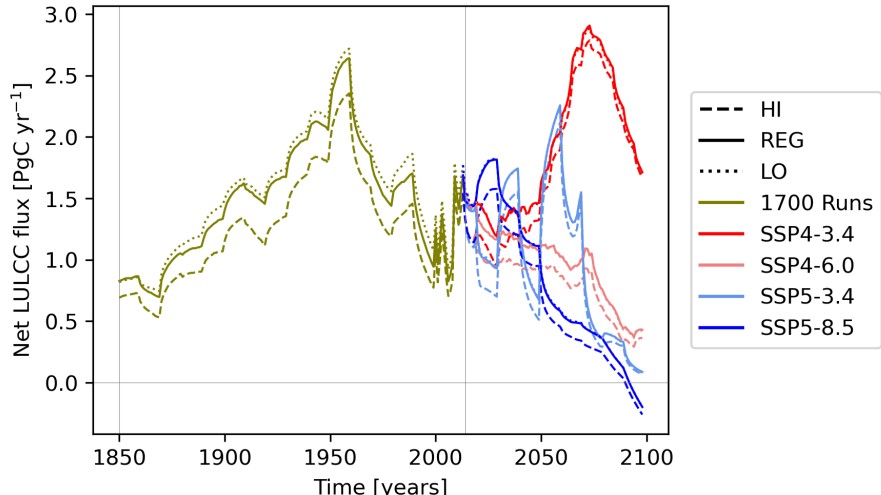

**Figure 1.** Global annual net LULCC flux for simulations with start year 1700 and HI, REG and LO LULCC scenarios of the LUH2 dataset (LO1700, REG1700 and HI1700). From 2014 onwards, each of the three historical simulations is continued with four different scenarios of future LULCC.

Feature (1) is not in conflict with a roughly symmetric uncertainty of harvest, which at first could be assumed to result in
equal difference in net LULCC flux between HI/REG and LO/REG. However, harvest on forested primary land, which is most important for the net LULCC flux, is similar between REG and LO (Fig. A2) and thus causes the similarity in net LULCC flux. Harvest on secondary land does not produce a net flux to the atmosphere if considered over a long time-period (total source is equivalent to total sink). From about 1800 onwards, less harvest on primary land can be observed in the HI LULCC estimate, slightly more in LO and the most in REG.
Feature (2) develops because the timescale of regrowth (sink of carbon flux, i.e. flux from atmosphere to land) is longer than that of clearing/harvest (source). The feature can be seen by comparing the orange and green crosses, representing the cumulative net LULCC flux for the period 1850–2014 in REG850 and REG1700 respectively, with the blue cross for REG1850 in Fig. 2.
Finally, feature (3) can be explained by the link between LULCC and the net LULCC flux. If one scenario has continuously
more LULCC than another, it will continue to produce a larger net LULCC flux and therefore no crossing points will occur. However, if the rate of LULCC varies differently with time in two scenarios, then the simulation with an initially larger amount of LULCC activities exhibits fewer transitions towards the end. This can for example happen when the setups have a similar beginning and end distribution of land cover, as is the case in the LUH2 dataset. The simulation with initially larger amount of LULCC activities produces an initially steeper increase of the cumulative net LULCC flux and a weaker increase towards the





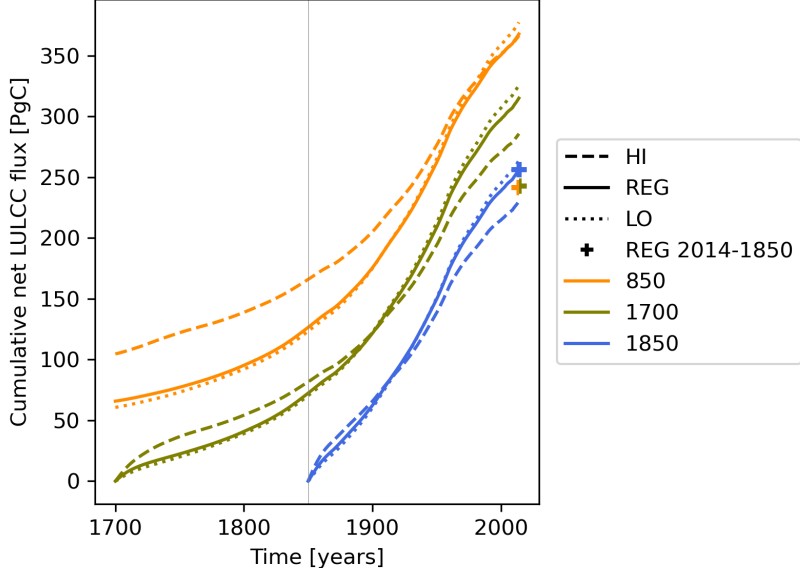

**Figure 2.** Global cumulative net LULCC flux since start of the individual simulations. Shown is the flux in the land-use scenario HI, REG and LO (compare lines with same colour) and the relevance of starting date (compare same linestyle across colours). The three crosses in 2014 represent the total cumulative net LULCC flux of the three REG experiments (REG850, REG1700 and REG1850) if the flux is only calculated for the period 1850–2014. Note that the orange and green cross overlap and are only partly visible.

end, which can potentially imply a crossing point. The likelihood for a crossing point can be enhanced if the two setups also start off with different land-cover distribution to amplify the change in rates or by reducing the considered period towards the end of the timeseries. The presence of harvest, which does not change the land cover distribution, and gross transitions masks this relationship. Still, the origin of these crossing points can, for example, be seen in Fig. A1c. Initially, and until about 1800, more natural land is converted to agricultural land in the HI scenario, but then a reversal of this trend relative to LO occurs (net

transitions not shown).

### 3.1.2 Comparison of components of uncertainty

As discussed in the previous section around Fig. 2, Fig. 3 similarly shows that the cumulative net LULCC flux in the LO scenario (filled circles) exceeds the values in the HI scenario (crosses). This is true both in the later part of simulations from 1850 (Fig. 3a, column LULCC) but also for the full simulation duration (Fig. 3b). Only the relative magnitude of the sensitivity

(spread along the $y$-axis for points with same $x$-axis base) of the cumulative net LULCC flux to LULCC and starting year of a simulation depends on the period considered. The cumulative net LULCC flux from 1850 (panel a) exhibits a reduced sensitivity to LULCC uncertainty with starting year 1850 (compare vertical spread of blue markers in LULCC-column) since the input data has smaller uncertainty in more recent years (Fig. A1). For the total cumulative net LULCC flux (panel b), uncertainties are small between LULCC setups in 850 (orange markers), since areas of cover fractions are most similar in the





beginning and the end. Sensitivity to the starting year (StYr, second column) is larger than to the LULCC estimate for the total cumulative net LULCC flux (Fig. 3b) but smaller towards the end of the timeseries (Fig. 3a). The similarity between the REG and LO experiments is apparent in both periods. Larger fluxes occur in runs from year 850 (column StYr) if the total cumulative net LULCC flux is considered (Fig. 3b), simply because of the longer model simulation (see Fig. B1). If only the common time period is considered (Fig. 3a, StYr), the largest estimates of the cumulative net LULCC flux are produced in

simulations from 1850.

In summary, the net cumulative LULCC flux is more sensitive to the LULCC uncertainty (22 % range in flux) and less sensitive to the starting year of the simulation (15 %) if the total net LULCC flux is compared over a common time period. Ranges of variability are not equally distributed around the reference simulations. For the standard setup REG1700, the influence of LULCC uncertainty (HI1700 and LO1700) is about three times larger than the sensitivity to StYr (REG850, REG1850). These

results are insensitive to the specific results of HI850, which shows a larger deviation from the LUH2 input dataset than the other experiments (Fig. A5).

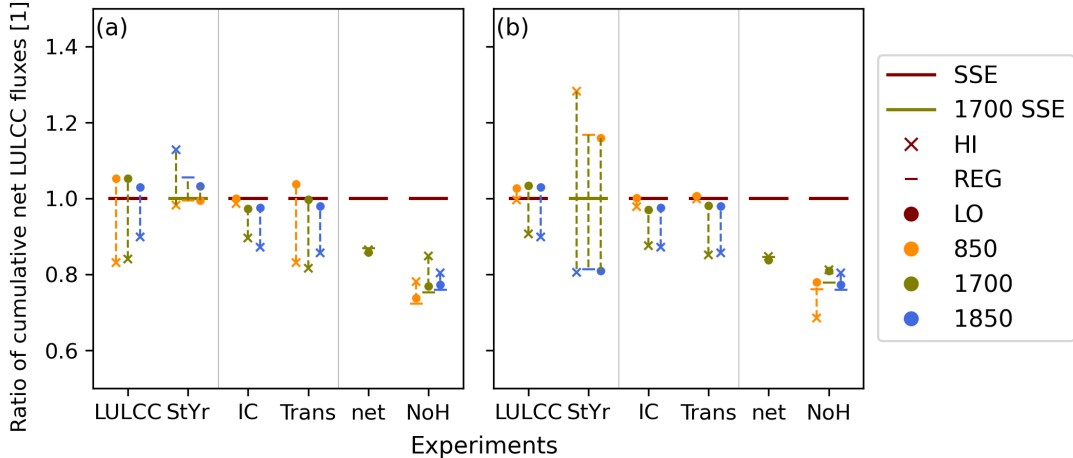

**Figure 3.** Comparison of global cumulative net LULCC flux between various simulations. Panel (a) shows the total flux between 1850 and 2014, normalised with a respective reference/standard sensitivity experiment (SSE, see also Table 3): REG*tt* for LULCC (first column), *xx*1700 for start year (second column), REG*tt* for IC and Trans experiments (third to fourth column) and *xxtt* for net and NoH experiments (fifth and sixth column). Here, *tt* (*xx*) means that the reference year (land-use scenario) is varying for each experiment in the respective column. Reference examples are REG1850 for HI1850 in the LULCC comparison, REG1700 for REG1850 in the StYr comparison and LO850 for LO850NoH in the NoH comparison. The color of the connecting lines represents the reference simulations. Panel (b) compares results from the same simulations but for the cumulative emissions during the whole simulation, thus for varying time periods.

The artificial sensitivity experiments IC and Trans reveal that the sensitivity to ICs (visible as the spread across LULCC estimates) increases more the later the simulation starts (Fig. 3, second and third column). Considering Trans, the temporal behaviour depends on the time periods considered: for the period 1850–2014 (Fig. 3a) the sensitivity of the cumulative net





LULCC flux decreases with later starting year while over the full respective time periods (Fig. 3b) the sensitivity increases with later starting year. These relative temporal characteristics can easily be explained by the divergence of land-use states from 850 to 1850 (Fig. A1a) and generally decreasing uncertainty of LULCC activities with time (i.e. visible in uncertainty of agricultural areas decreasing after 1700, Fig. A4). As the extent of agricultural areas increases after 1700, the reduced amount of agricultural land in IC and Trans experiments (Fig. A4) implies fewer transitions to crop and pasture. This likely explains

why for simulations with start year 1700 all sensitivity experiments exhibit lower or at most equal cumulative net LULCC flux than the reference scenario REG1700. The temporal evolution of the cumulative net LULCC flux from IC and Trans experiments (not shown) confirms that crossing points originate from the variability of LULCC activities, because they only occur in Trans and not in IC experiments. The sensitivity of the cumulative net LULCC flux due to uncertainties in transitions (Trans) is larger (up to 21 %) than due to initial conditions (IC) (between a few percent to 11 %). Note that the sensitivity of net

LULCC flux to IC and Trans is not expected to be additive to the total sensitivity combining initial conditions and transitions (the LUH2 input in columns LULCC and StYr) due to several reasons: the biosphere and soil stocks are not in equilibrium at the end of a simulation (e.g. Stocker et al., 2011) and, as already mentioned, the simulations start and end with different land-cover distributions.

The sensitivity of the cumulative net LULCC flux to net versus gross transitions (Fig. 3a, fifth column, about 13 % for

REG1700) is of similar order of magnitude as from the starting year of a simulation (StYr). Furthermore, all setups roughly exhibit the same ratio of net LULCC flux with net or gross transitions (Fig. 3a and b).

Neglecting harvest has a larger impact on the cumulative net LULCC flux (up to 28 % reduction) than the total sensitivity to the uncertainty of LULCC (Fig. 3a, sixth column). Harvest is also the main driver of the asymmetry between cumulative net LULCC fluxes from HI/REG/LO scenarios after 1850 (panel a). For the cumulative net LULCC flux from 1850 (Fig. 3a),

omitting harvest causes the least reduction in HI and the most in REG. This result can be explained by the relative amount of deforestation on forested primary land. Considering the total cumulative net LULCC flux (Fig. 3b), neglecting harvest and its uncertainty means that the sensitivity to total LULCC uncertainty for simulations started in 1700 and 1850 is considerably reduced. Interestingly, the reduction in cumulative net LULCC flux is largest in HI850NoH if considering the whole simulation (Fig. 3b) but from 1850 (Fig. 3a), LO850NoH and REG850NoH show the largest reduction by omitting wood harvest.

The main conclusions from the comparison of the given estimates of LUH2 LULCC uncertainty with other sources of sensitivity to the cumulative net LULCC flux over the period 1850–2014 are: 1. the net LULCC flux is most sensitive to accounting for wood harvest (decrease of up to 28 % without wood harvest), 2. the sensitivity due to LULCC uncertainty (22 %) is larger than from StYr (15 %), 3. the cumulative net LULCC flux is similarly sensitive to StYr and net versus gross transitions (13 %) and 4. later starting years reduce sensitivity to uncertainties in LULCC activities and increase sensitivity to

uncertainties in initial conditions.





**Table 3.** Overview of sensitivity experiments for Fig. 3 and 4. Combine row and column to one experiment setup (note that LULCC and StYr do not modify the setup, but IC, Trans, net and NoH do) and then find the corresponding reference setup in the table. If several experiments are given the ordering is the same as in the column header.

|  | LO850/ REG850/ HI850 | LO1700/ REG1700/ HI1700 | LO1850/ REG1850/ HI1850 |
|---|---|---|---|
| LULCC | REG850 | REG1700 | REG1850 |
| StYr | LO1700/ REG1700/ HI1700 |  | LO1700/ REG1700/ HI1700 |
| IC | REG850 | REG1700 | REG1850 |
| Trans | REG850 | REG1700 | REG1850 |
| net | LO850/ REG850/ HI850 | LO1700/ REG1700/ HI1700 | LO1850/ REG1850/ HI1850 |
| NoH | LO850/ REG850/ HI850 | LO1700/ REG1700/ HI1700 | LO1850/ REG1850/ HI1850 |

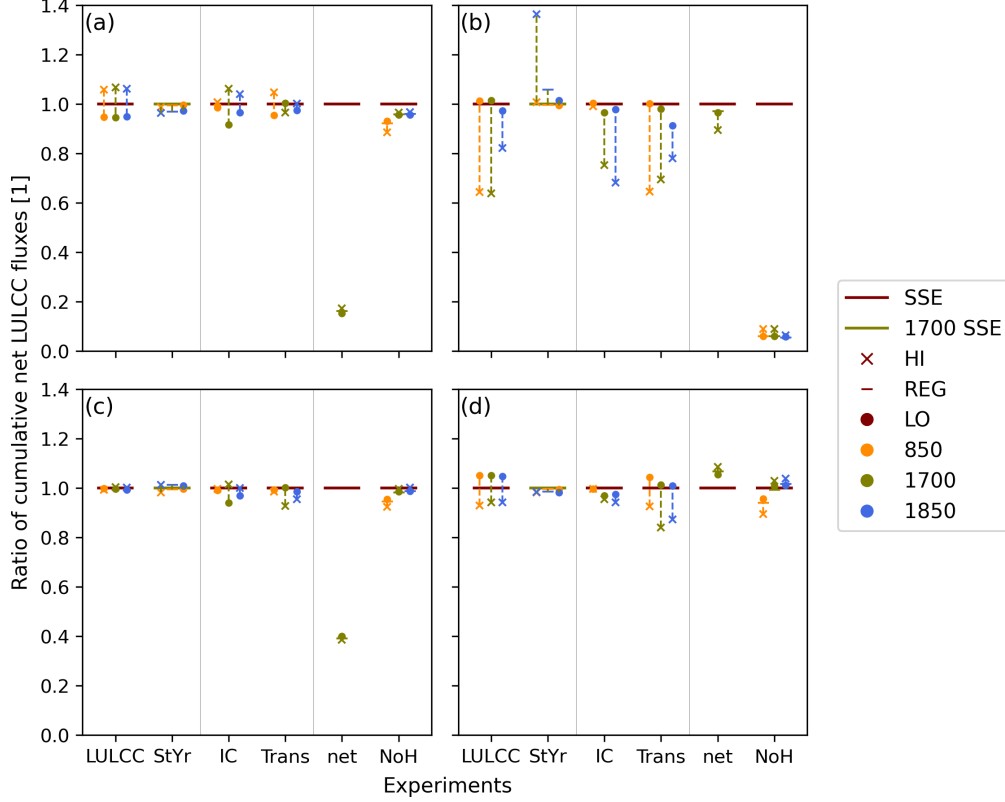

**Figure 4.** Global cumulative net LULCC flux as in Fig. 3a for the period 1850–2014 but separated by LULCC activity: (a) abandonment, (b) harvest, (c) crop expansion and (d) pasture expansion. Normalisation is done individually for each LULCC activity. SSE refers to the standard sensitivity experiment.



### 3.1.3 Impact of uncertainties of different LULCC activities on net LULCC flux

The relative ordering of the impact of uncertainties of LULCC activities on the net LULCC flux (vertical spread of experiments in the LULCC column) is: Harvest contributes the most (up to 40 % relative to the reference setup, Fig. 4b), uncertainties in abandonment and pasture expansion cause approximately equal shares of sensitivity in total net LULCC flux (about ±10 %,

Fig. 4a and d), larger than LULCC uncertainty for crop expansion (Fig. 4c). Note that abandonment is a carbon sink, thus values larger than 1 indicate a stronger sink.

    Sensitivity of the cumulative net LULCC flux to harvest is mainly found for HI setups (LULCC column) and any LULCC simulation started in 1850 (StYr column). Uncertainty of wood harvest also explains why REG850 and LO850 produce the same amount of harvest emissions until 1700 (Fig. 2), although their harvest transitions show different extent. Harvest on

primary land is mostly different from REG in the HI setup (Fig. A2), in line with similar net LULCC flux estimates in REG and LO. Both harvest and pasture expansion exhibit larger cumulative net LULCC flux in LO than HI experiments (Fig. 4b and d), while the opposite is true for abandonment (Fig. 4a) and crop expansion shows minimal differences between the two experiments (Fig. 4c). Global crossing points in total net LULCC flux (Fig. 2), corresponding to larger cumulative net LULCC flux in LO than HI experiments, are thus due to pasture expansion and harvest. The LULCC activity showing the best agreement

between the three LULCC scenarios is crop expansion (Fig. 4c): results of HI and LO experiments, as well as with StYr 850 and 1850 deviate little from REG1700 (columns LULCC and StYr). Thus most of the comparably large uncertainty of crop transitions presented in Fig. A2 is associated with shifting cultivation and does not impact the net LULCC flux.

    The sensitivity of the net LULCC flux to the uncertainty from pasture expansion (Fig. 4d) is larger from transitions (Trans, fourth column) than from initial conditions (IC, third column). This can be explained by the fact that the agricultural area (Fig. A4) shows a larger spread in Trans (red line) than IC experiment (blue line) towards the end of the timeseries, which is larger

than between the HI, REG and LO experiments (green lines) in both setups. Similarly to findings from the main experiments, the order of IC experiments for harvest and pasture (which show largest net LULCC flux in the LO scenario) is opposed to crop and abandonment (largest net LULCC flux in the HI scenario), which is likely related to shifting cultivation. A reduction of the cumulative net LULCC flux in IC and Trans experiments initialised in 1700 or 1850 is both due to reduced contribution

from harvest and pasture (only IC) and the opposite ordering of LULCC experiment in crop and abandonment contributions.

    The use of net instead of gross LULCC forcing leads to the largest decrease in the net LULCC flux components from abandonment and crop expansion (80 % and 60 %, respectively), which can be expected due to shifting cultivation. Net transitions slightly decrease the contribution from harvest and increase the contribution from pasture expansion to the net LULCC flux, both by about 10 %. The latter is most likely caused by pasture expansion occurring on previously less intensively

used land with thus larger carbon stocks. If wood harvest is neglected, all other LULCC activities approximately produce the same spread of cumulative net LULCC flux, i.e. the ratio of a simulation with and without wood harvest is about one (see Table 3 for reference simulations). Only the net LULCC flux from simulations in starting 850 is slightly reduced. Note that in the experiments without harvest, the cumulative net LULCC flux from harvest is not zero because a small contribution of transitions from primary to secondary land due to rangeland expansion is counted as harvest.





The analysis of the contributions from the four LULCC activities to the total net LULCC flux sensitivity reveals: 1. LULCC uncertainty from harvest causes largest sensitivity in the cumulative net LULCC flux, followed by equal contributions from abandonment and pasture and negligible sensitivity due to crop uncertainty. The sensitivity of the cumulative net LULCC flux is measured relatively to the flux from REG1700. For harvest the sensitivity is asymmetric, i.e. the net LULCC flux due to harvest in the HI scenario deviates further from REG than in the LO scenario. 2. Years of equal emissions in different sensitivity experiments before the end of the simulation (crossing points) in the global net LULCC flux evolution are likely caused by pasture expansion and wood harvest. 3. Uncertainties in wood harvest cause large sensitivity to starting year of the simulation (StYr), as well as to initial conditions (IC) and transitions (Trans) in the artificial LULCC experiments.

### 3.1.4 Regional variations of uncertainty

Europe, Asia and Africa exhibit the largest sensitivity of cumulative net LULCC flux to LULCC uncertainties in the REG, HI and LO simulations starting in 1700 (Fig. 5). In most regions, HI1700 produces a smaller cumulative net LULCC flux than REG1700 and the cumulative flux is generally larger in LO1700 than REG1700. However, there are large coherent areas over Central and North America and Northern Europe/Asia with reduced cumulative net LULCC flux in LO1700 compared to REG1700.

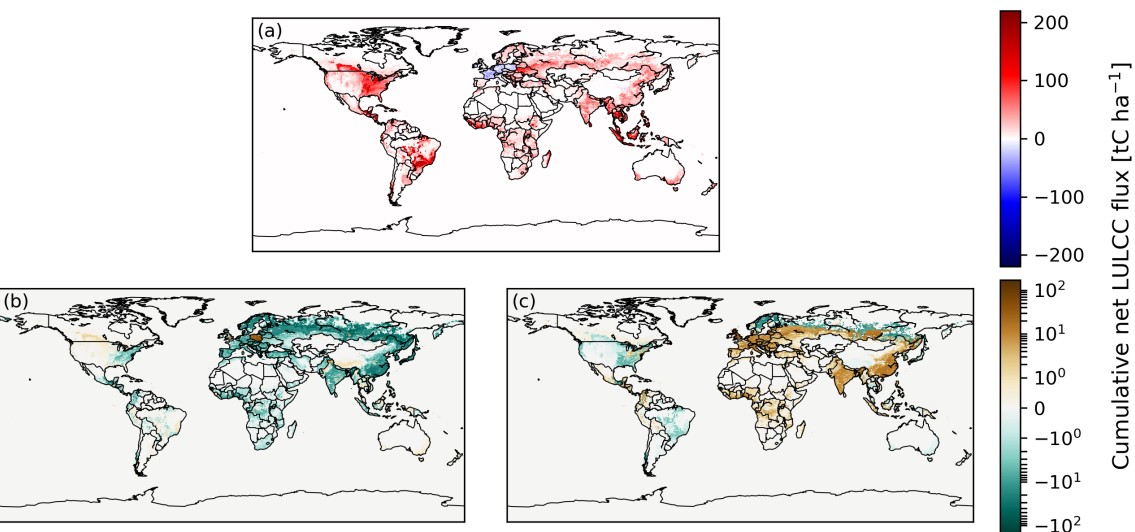

**Figure 5.** Cumulative net LULCC flux for the period 1850–2014 from REG1700 (a) as well as the the difference HI1700-REG1700 (b) and LO1700-REG1700 (c).



Some regions with reduced emissions in the HI scenario, like Poland and South-East Asia, correspond to regions where
fewer transitions of the LUH2 input data are used (Fig. A5), which is further enhanced in the HI-REG comparison.

Further division by LULCC activity is discussed in the following and shown in the Supplementary Material (see Fig. B2).
Cumulative net LULCC flux estimates are most sensitive to harvest uncertainties, mainly over Northern Europe, Northern Asia
and South-Eastern Asia (China and North-Eastern India). Components of the cumulative net LULCC flux due to uncertainty
of crop expansion and abandonment follow the pattern of shifting cultivation in the Tropics which means that the sensitivity to
uncertainties in abandonment and crop are balanced with opposite sign. Largest sensitivity of the cumulative net land-use flux to
LULCC using net transitions is present over Europe from abandonment and over India and South-East Asia from uncertainties
in crop transitions. The sensitivity of the net LULCC flux to uncertainties of pasture and overall uncertainty of LULCC over
Oceania is relatively small. Interestingly, the cumulative net-land use change flux over Oceania is larger in HI1700 rather than
LO1700 because few transitions occur before 1700 so that basically all transitions are captured in the analysis period.

## 3.2    How does past uncertainty impact future scenarios?

### 3.2.1    The current state

Next, we want to analyse the magnitude of legacy emissions at the end of the historical simulations in 2014 and how much
they are affected by past LULCC uncertainty. The magnitude of the annual net LULCC flux is determined by the size of the
disequilibrium pools, which aggregate information of past LULCC events. If these disequilibrium pools are similar between
two setups in a given year and the upcoming LULCC events are identical, then the annual net LULCC flux in the following
years will be similar as well.

In 2014 the annual net LULCC flux is 1.7 PgC yr$^{-1}$ in REG1700 (Fig. 6). Neglecting wood harvest (NoH) or only using
net transitions (net) leads to three times larger deviations from the reference (see Table 3) than LULCC uncertainties (first
column) and reduces the net LULCC flux at most to about 1.1 PgC yr$^{-1}$. The 5–10 % sensitivity of the net LULCC flux to
LULCC uncertainties (about 1.55 to 1.75 PgC yr$^{-1}$) can mainly be explained by the uncertainty of transitions. Almost no
sensitivity of the net LULCC flux to the starting year of the model simulations remains and the relative relationships between
the nine main simulations are similar as discussed for cumulative net LULCC flux estimates (Fig. 3). LULCC differences still
modulate annual net LULCC flux estimates throughout the 20$^{th}$ century (Fig. B3) and the largest variability of net LULCC flux,
about $\pm 0.1$ to 0.3 PgC yr$^{-1}$ is due to uncertainties in harvest and abandonment. In 2014, the largest impact of the remaining
differences is due to harvest (about $\pm 0.05$-0.1 PgC yr$^{-1}$).

### 3.2.2    Estimates of future emissions

The extensions of the twelve scenario simulations as a continuation of the three historical simulations with starting year 1700
are shown in Fig. 1. The underlying area changes are presented in Fig. A3 and the attribution of emissions to different land-use
histories is shown in the supplementary material (Fig. B4). Table 4 provides the annual and cumulative emissions of the twelve
scenario simulations. For each experiment, the first number is the SSP setup and the second the prescribed RCP value.





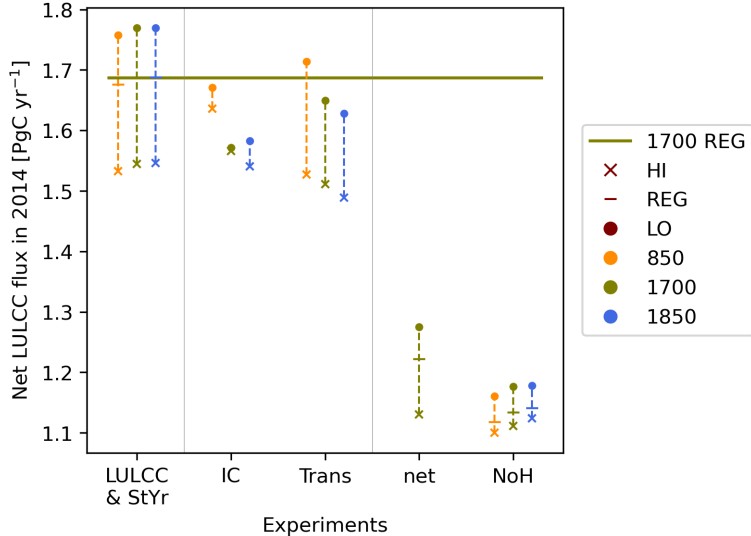

**Figure 6.** Global annual net LULCC flux in 2014. Although the overall layout is as in Fig. 3 the *y*-axis is not scaled by a reference simulation but presents the total net emissions in 2014. Note that the experiment groups LULCC and StYr are now combined as the presented values show the absolute net LULCC flux.

Land use in the baseline SSP4 scenario (SSP4-6.0) leads to a relatively steady decline of the net LULCC flux over the 21st century. More stringent mitigation policies (RCP3.4) result in an initial plateau of the net LULCC flux up to the middle of the 21st century, followed by a peak in the second half of the 21st century of similar magnitude as the maximum in the 1950s. This peak is mainly caused by crop expansion and a reduced sink from abandonment connected to a reduction of secondary land
area from about 2050 (Fig. B4a, c).

The baseline SSP5 scenario (SSP5-8.5) on the other hand starts off with a minor maximum of the net LULCC flux which is followed by a declining estimate. The initial peak in SSP5 is mainly caused by pasture expansion and wood harvest (Fig. A3); the evolution of secondary land and cropland is similar as in the SSP4 baseline, but less area is used for pasture. Overall, the net LULCC flux in 2099 is lower than in SSP4-6.0 by about 0.6 PgC yr$^{-1}$. In the alternative 3.4OS scenario, which differs
from the SSP5 baseline mainly after 2040, a secondary peak after around 2050 is present, mainly caused by crop expansion over pasture.

Remaining sensitivities to LULCC uncertainties in future scenarios are due to harvest (Fig. B4) and decrease towards the end of the 21st century but do not reach zero in 2099. These uncertainties in harvest also explain why the remaining spread of net LULCC flux is larger in HI than LO, similar to the historical period.

The estimates of annual net LULCC flux estimates in 2099 (Table 4) indicate a reduction of sensitivity to LULCC uncertainties from $\pm 0.15$ PgC yr$^{-1}$ to between $\pm 0.07$ PgC yr$^{-1}$ and $\pm 0.02$ PgC yr$^{-1}$, respectively, for SSP4-6.0 and SSP5-3.4OS. Note that with an accuracy of the net LULCC flux of 0.1 PgC yr$^{-1}$, a difference in the future scenarios due to LULCC uncertainty





**Table 4.** Annual net LULCC flux [PgC yr$^{-1}$] in 2014 for historical simulations starting in 1700 and in 2099 for four SSP scenarios. For the REG experiments, the cumulative net LULCC flux of the future scenarios (2014-2099) is included in brackets [PgC].

|      | historical | SSP4-3.4   | SSP4-6.0  | SSP5-3.4OS | SSP5-8.5  |
|------|------------|------------|-----------|------------|-----------|
| HI   | 1.54       | 1.67       | 0.36      | 0.07       | -0.26     |
| REG  | 1.69       | 1.71 (162) | 0.43 (82) | 0.09 (87)  | -0.2 (69) |
| LO   | 1.78       | 1.68       | 0.42      | 0.08       | -0.2      |

only remains in SSP5-8.5. The difference in net LULCC flux between historical LULCC uncertainty setups for individual scenarios in 2099 is about 50 % of their spread in 2014. This reduction occurs the most in RCP3.4 scenarios. In both SSP4 and
SSP5, the cumulative net LULCC flux is larger with lower RCP value (values in brackets in Table 4). The impact of the initial uncertainty is thus further reduced, relative to the magnitude of the net LULCC flux in 2099, if followed by a larger cumulative net LULCC flux. Scenarios with reduced radiative forcing due to increased mitigation action (RCP3.4) produce increased cumulative net LULCC fluxes over the 21$^{st}$ century, since fossil fuel emissions are substituted partly by energy from biofuel (Hurtt et al., 2020). This biofuel production causes additional cropland expansion and thus leads to net LULCC fluxes from
LULCC (see Fig. B4c). Still, the total carbon emissions are expected to be larger in the baseline than the RCP3.4 scenarios.

### 3.3 Evaluation against previous studies

Our baseline scenario REG1700 exhibits a cumulative net LULCC flux of 242 PgC for the period 1850–2014. The sensitivity range due to LULCC uncertainty and starting year is about 22 % for comparable setups. In the nine main experiments the cumulative net LULCC flux is at least 201 PgC (HI850) and at most 264 PgC (LO1850). The relative change due to neglecting
gross transitions is similar across LULCC setups and for REG1700net the cumulative net LULCC flux is reduced to 211 PgC. Wood harvest causes the largest sensitivity in the cumulative net LULCC flux (the flux in REG1700NoH is 175 PgC).

  The sensitivity results presented here are limited by the fact that i) initial and final areas of land cover are not the same in the different experiments, ii) the disequilibrium pools are not the same in 2014 because timescales of harvest and regrowth differ (no committed emissions), iii) the uncertainty range of LUH2 is not exhaustive but represents known uncertainties (un-
intuitively the known uncertainty is larger in data-rich regions), and iv) BLUE does not use 100 % of the suggested transitions from the LUH2 input dataset.

  Point iv) mostly affects usability of results from experiment HI850. Considering the whole time period, HI850 produces results between HI850NoH and the setup suggested by the LUH2 dataset, but closer to the latter. As differences of primary land area in 2014 between the LUH2 dataset and the BLUE experiments are otherwise uniform across LULCC scenario
experiments, the qualitative properties of the results will be valid also if accurately using the whole dataset.

  Over the period 1850–2014 the cumulative LULCC flux as determined by GCB2019 (Friedlingstein et al., 2019) is $195\pm 60$ PgC, compared to $400\pm 20$ PgC from fossil fuels. The baseline scenario is thus included in the GCB2019 uncertainty range; the sensitivity range of the cumulative net LULCC flux due to LULCC uncertainty is smaller than the uncertainty in GCB2019





but the sensitivity due to inclusion of wood harvest is of similar magnitude. However, towards the end of the historical time

series, the sensitivity of the net LULCC flux to LULCC uncertainty and to all other parameters is somewhat smaller than the uncertainty presented in Friedlingstein et al. (2019) of $1.5\pm0.7$ PgC yr$^{-1}$ (2008–2019). Around the baseline estimate of 1.7 PgC yr$^{-1}$ (REG1700) LULCC adds asymmetrically about $\pm0.15$ PgC yr$^{-1}$ and without harvest or gross transitions the net LULCC flux in 2014 is reduced by 0.6 PgC yr$^{-1}$. The importance of LULCC uncertainty on net LULCC flux decreases with time and therefore is more relevant for the cumulative net LULCC flux than for the annual value in 2014.

Modelling studies of Stocker et al. (2014) and Arneth et al. (2017) agree that the contribution from shifting cultivation, i.e. gross transitions, and wood harvest are of similar magnitude and increase the net LULCC flux, though both studies base their estimates on different time periods and are therefore not necessarily comparable. Stocker et al. (2014) quantify the contribution to the total net LULCC flux at 19 % each from wood harvest and shifting cultivation over the period 2000–2009, which can be added to the base value of 1.2 PgC yr$^{-1}$. In Arneth et al. (2017) the estimate of 30 % increase to the base value of $119\pm50$ PgC

due to both shifting cultivation and wood harvest is obtained with 7 DGVMs and valid for the period 1901–2014. Wilkenskjeld et al. (2014) find a reduction of the cumulative net LULCC flux by 38 % if shifting cultivation is not considered (1850–2005). The estimates found here with BLUE and the LUH2 dataset (a 13 % decrease by neglecting shifting cultivation and 28 % decrease by neglecting wood harvest) are thus comparable in magnitude to previous studies.

     These results are also largely consistent with the findings of Hurtt et al. (2011), in which the contribution of shifting cul-

tivation and wood harvesting were the model factors that the simulation output, in terms of the net LULCC flux, was most sensitive to. In comparison with Hurtt et al. (2011) it can be noted that sensitivities might look different in other metrics like forest age or area. Although the spatial and temporal representation of these processes has been significantly improved in LUH2 (vs. LUH1), the choice of whether or not to include these processes in DGVM simulations is still a large contributor to the overall uncertainty in LULCC fluxes. However, assuming that the sensitivity in net LULCC flux from one LULCC dataset

with uncertainties (based on the LUH2 dataset, presented here) is similar to the comparison of two LULCC datasets (Gasser et al., 2020), results in Gasser et al. (2020) point towards even larger contributions from e.g. uncertainties in carbon densities (both spatially and temporally).

## 4   Conclusion

This study investigates the impact of LULCC uncertainties compared to other common uncertainties in modelling of LULCC

fluxes with the bookkeeping model BLUE, like the representation of wood harvest and shifting cultivation.

     We show that the sensitivity of the net LULCC flux to the uncertainty of LULCC based on the LUH2 datset is not negligible and may explain part of the large uncertainty range of DGVMs as part of the GCB (Friedlingstein et al., 2019), since LULCC processes are captured with varying comprehensiveness (see Table A1 in Friedlingstein et al., 2019). The LULCC uncertainty has comparable impact on the cumulative net LULCC flux to including harvest and gross transitions while its impact on most

recent annual estimates is about three times smaller. For the starting years presented here (850, 1700 or 1850), the spread in cumulative net LULCC flux is about the same order as from including gross transitions but can be neglected for annual fluxes in





recent years. This means that it is of little importance for estimates of the net LULCC flux over recent years when a simulation was started but it is important for cumulative fluxes, with relevant implications for comparisons of the GCB and CMIP6 model simulations. However, not accounting for gross transitions and wood harvest, as is sometimes still the case in DGVMs, can

cause even larger differences between model estimates. Finally, it should be noted that the two alternative LULCC scenarios (low and high land-use scenario) produce relatively smaller or larger estimates of the net LULCC flux than the LUH2 baseline scenario depending on the time period considered.

Furthermore, the difference in net LULCC flux between high and low land-use scenarios is expected to be larger in DGVMs than in a bookkeeping model as they are influenced by a higher $CO_2$ concentration exposure via the loss of additional sink

capacity. In DGVM simulations, a higher $CO_2$ exposure will most likely lead to larger vegetation and soil carbon stocks in the 20$^{th}$ century in low simulations as compared to high land-use simulations. The increasing number of transitions in the 20$^{th}$ century in the low land-use simulations will thus increase the difference in emissions between the two alternative scenarios. Another difference that can influence results comparing bookkeeping models and DGVMs is that the former approach uses constant (present-day) carbon densities while DGVMs work with variable carbon densities which respond to environmental

conditions. Nevertheless, the results presented here provide a reference for comparisons with the upcoming CMIP6 model simulations.

## 5 Acknowledgements

We acknowledge use of the LUH2 data, which was obtained from https://luh.umd.edu/. A part of the BLUE model simulations was executed on the Linux Cluster hosted by the Leibniz-Rechenzentrum in Munich. JP was supported by the German Research

Foundation's Emmy Noether Program (PO 1751/1-1). RG acknowledges support from the European Commission through Horizon 2020 Framework Programme (VERIFY, grant no. 776810).





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



## Appendix A: The LULCC forcing

### A1   Preparation of LUH2 data for BLUE

Land-use areas and LULCC activities of the input dataset LUH2 are aggregated to fit the input requirements of BLUE, namely the four land-cover types primary land, secondary land, pasture and crop. LUH2 primary forested and primary non-forested land areas are combined to primary land in BLUE. Rangeland is attributed depending on the forest/non-forest map (variable fstnf of file http://gsweb1vh2.umd.edu/LUH2/LUH2_v2h/staticData_quarterdeg.nc) from LUH2 to either pasture (if forest) or secondary land (non-forest). LUH2 secondary forested and secondary non-forested land areas are combined with the contribution from rangeland to secondary land areas. BLUE pasture consists of managed pasture, the rangeland contribution and urban area. Finally, LUH2 C3/C4 annual/ perennial crop are combined as cropland in BLUE.

Transitions between land-use types are aggregated in the same way as the land-use types themselves. In addition, wood harvest is used by means of harvested area (as opposed to the alternatively available harvested biomass). Transitions from primary to secondary land which are not associated with wood harvest are still accounted as part of wood harvest. Since areas and transitions from the LUH2 dataset refer to fractions of the total gridcell, we scale them down with a map of total vegetation cover in each gridcell (Pongratz et al., 2008).

Since harvest is provided in the LUH2 dataset based on the cover type (forest or non-forest), transitions are not used in BLUE when the cover type does not match. Fig. A5 shows the impact of these neglected transitions in terms of difference in primary land between the LUH2 input dataset and BLUE in 2014, i.e. at the end of the historical simulation period. Differences are reduced for later starting time (g) and then the spread between the three LULCC scenarios is also reduced (e.g. h and i). Largest differences between BLUE and the LUH2 dataset occur in HI850 (b). Transitions on primary non-forested land are mainly neglected in Europe, especially Poland, the Middle East, India and in western Africa (not shown). Larger coherent regions affected by missing transitions on primary forested land are the Tibetan plateau and extended areas over Russia. The difference in global primary land area between the LUH2 dataset and BLUE is mostly below 5 %, for all simulations starting in 1850 less than 2 %, and only HI850 exhibits a deviation of 12 %. These neglected transitions on primary land can induce omissions of follow-on LULCC activities if the required land-use type is not available since transitions are only executed in BLUE if the from-type is present. Due to this rule it is also not possible that the area fraction in a grid cell exceeds 100 % due to previously neglected transitions.

### A2   Properties of the LULCC dataset

The properties of the LULCC LUH2 dataset (Hurtt et al., 2019a, b) are presented in Hurtt et al. (2020) and are briefly discussed here with modifications for the analysis with BLUE and to provide a basis for the following sensitivity analysis. Properties of land-use areas and LULCC activities are first discussed in the baseline scenario (here called REG) and then differences in the high (HI) and low (LO) LULCC scenarios are compared.





The amount of secondary and agricultural land in 850 is small compared to primary vegetation (Fig. A1a,b, less than about 1000 and 200 Mio. ha, respectively, compared to more than 8000 Mio. ha of primary land). From around 1700 the area of
645 agricultural land expands more rapidly and from around 1850 the same is true for secondary land (Fig. A1c and d, respectively). Abandonment and crop expansion (Fig. A2a) are of similar magnitude due to shifting cultivation dominating gross LULCC (not shown), especially until 1750. From 1300 onwards, and for most of the time series, these two LULCC activities affect roughly the same area as wood harvest, though wood harvest exhibits larger temporal variability. Pasture expansion and harvest on primary forested land are only relevant from around 1700 onwards and affect less area than the other LULCC activities.

The uncertainty of agricultural area is largest at the beginning of the timeseries (Fig. A1b) and decreases with time. In 850 the uncertainty around the baseline scenario is about 50 % for pasture and crop area, of which 1 % remain in 2014 (Fig. A1b). The uncertainty in secondary land is about 50 % in 850 (Fig. A1a). This initial uncertainty of secondary land is due to division of rangelands into secondary land and pasture for BLUE and is accounted to rangelands in the LUH2 data. Thus, the same total uncertainty is present in the LUH2 dataset and the data prepared for BLUE. It is important to note that the historical
scenarios (HI, REG and LO) neither start nor end with the same area distribution. The transitions (Fig. A2b-e) show largest uncertainty in wood harvest, with a small contribution from wood harvest on primary land. Compared to total wood harvest the contribution of harvest changing cover type from primary to secondary land is relatively small. Although total harvest biomass is designed to be equal across scenarios after 1920 (Hurtt et al., 2020), this is not true for harvested area, since harvested area is derived such that the demanded harvested biomass can be fulfilled. Since the other LULCC activities influence the available
biomass, more or less area might be required in order to fulfill the harvested biomass demand. Increased uncertainties in crop and abandonment before 1850 are largely related to uncertainties about the magnitude of shifting cultivation and the extent of agricultural areas described in the HYDE dataset.

The baseline SSP5 scenario (SSP5-8.5) captures conditions of high levels of fossil fuel use, increasing global food demand and therefore increasing cropland area (about 20 % increase from 2010 to 2100, Fig. A3). At the same time, primary land
area is reduced to about 74 % of its original extent and secondary land area is steadily increasing at a total of about 22 %. The alternative scenario SSP5-3.4OS is an overshoot scenario which mainly differs from the baseline scenario after 2040. The cropland area is increased by 50 % from 2010 to 2100, mainly by cultivating cropland which was previously used as pasture. The evolution of primary and secondary land areas is similar to the baseline.

The baseline SSP4 scenario (SSP4-6.0) represents an evolution of progress with high agricultural productivity and envi-
670 ronmental policies (reduced deforestation, re- and afforestation, ...) in high-income countries and the opposite in low-income countries. The alternative scenario SSP4-3.4 is based on more stringent mitigation policies, e.g. a larger carbon price. Similar to SSP5, the increase in cropland area is larger in the lower RCP scenario, namely 14 % and 80 % respectively for RCP6.0 and RCP3.4 between 2010 and 2100. Both scenarios exhibit stronger decline in primary land area than SSP5 and the additional expansion of cropland in RCP3.4 goes along with reduced extent of secondary land.





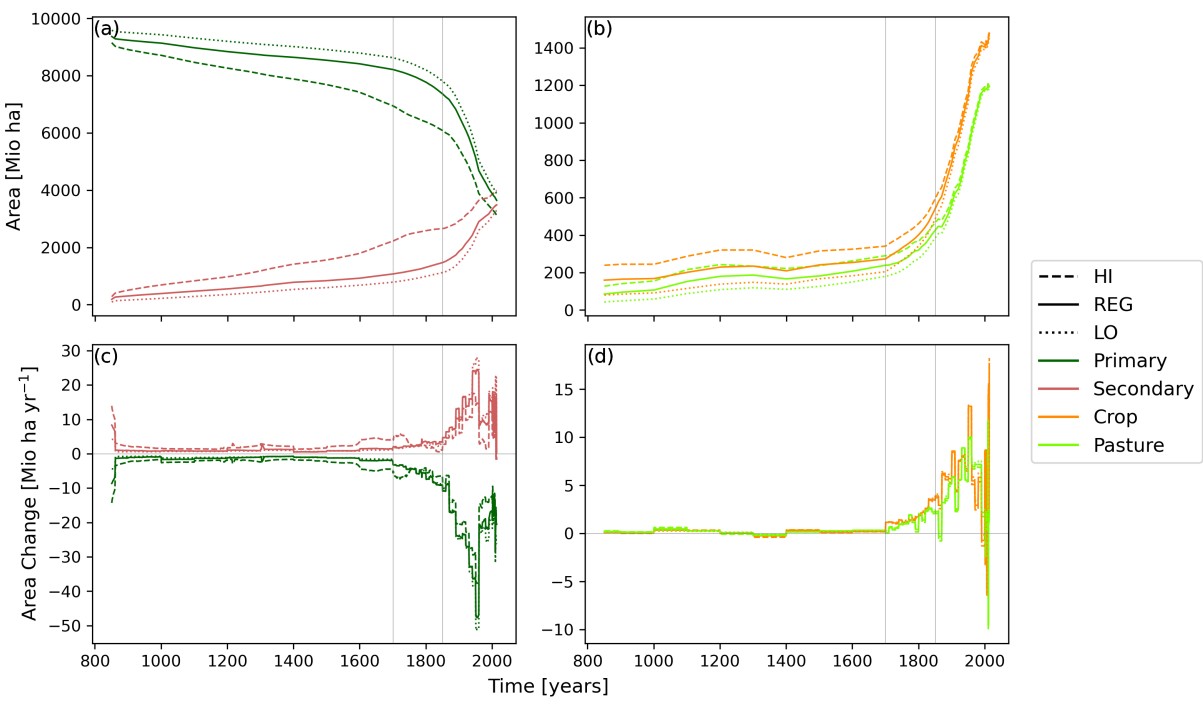

**Figure A1.** Global areas of the four BLUE land-cover types primary land, secondary land, crop and pasture based on the aggregated LUH2 input data (a, b) and their temporal net change (c, d). Panels (a) and (c) show natural vegetation, (b) and (d) agricultural area. Note the different *y*-axis ranges. HI, REG and LO corresponds to LUH2 LULCC estimate high, baseline and low, respectively.



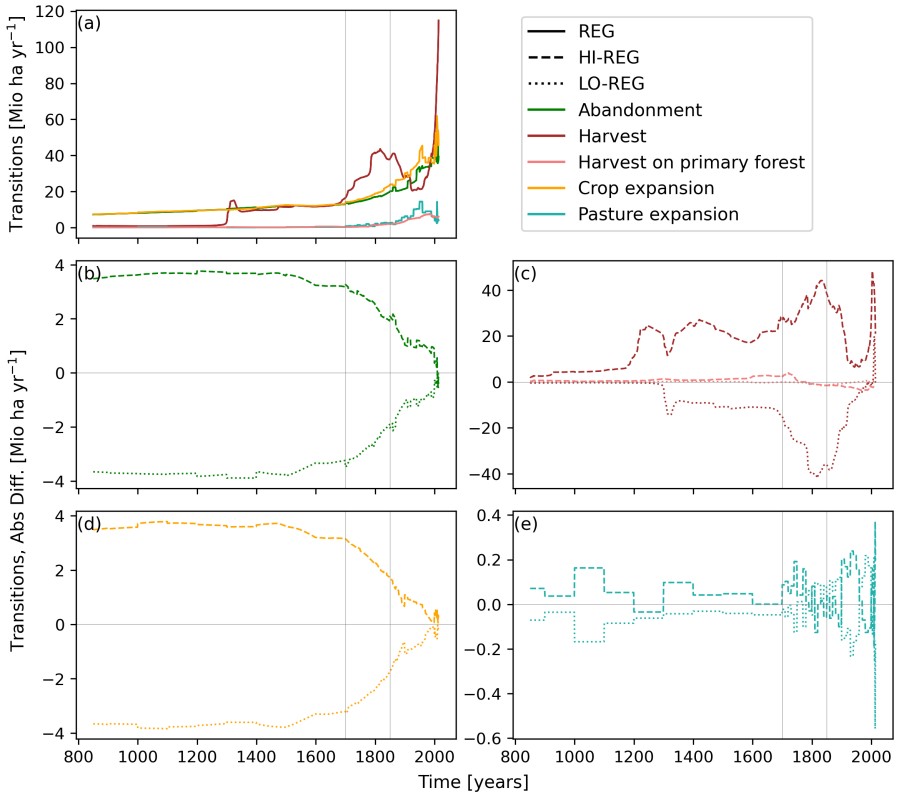

**Figure A2.** Global gross transitions based on LUH2 baseline scenario (panel a, REG) and absolute difference of high (HI) and low (LO) land-use estimate compared to baseline LUH2 setup (b-e). For harvest (panel c) the sub-transition of harvest on primary forest is shown as well. Note the different *y*-axis ranges.

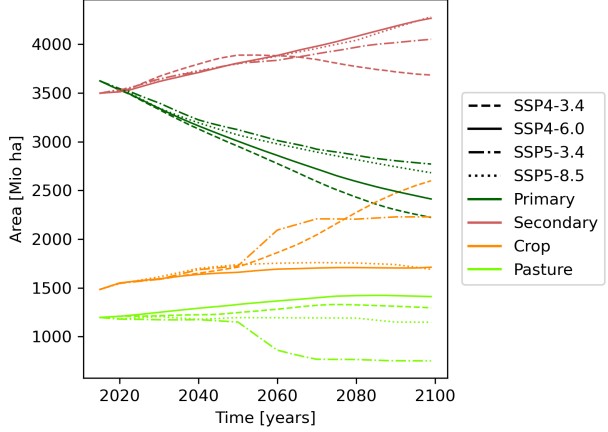

**Figure A3.** Global areas of the four BLUE land-cover types based on the LUH2 dataset in four future scenarios described in the text.





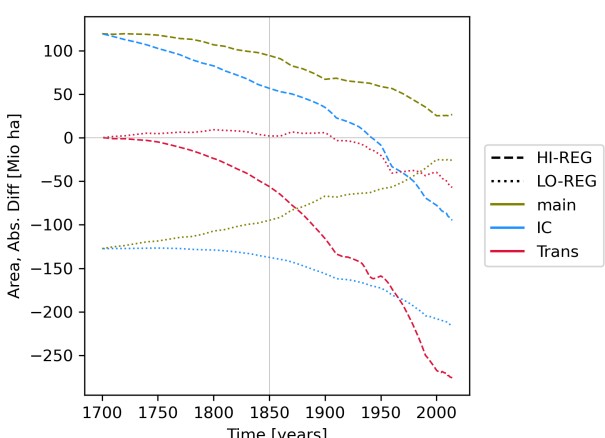

**Figure A4.** Differences in global total agricultural area in BLUE, also including results from initial conditions (IC) and transitions (Trans) sensitivity experiments (see Table 2).




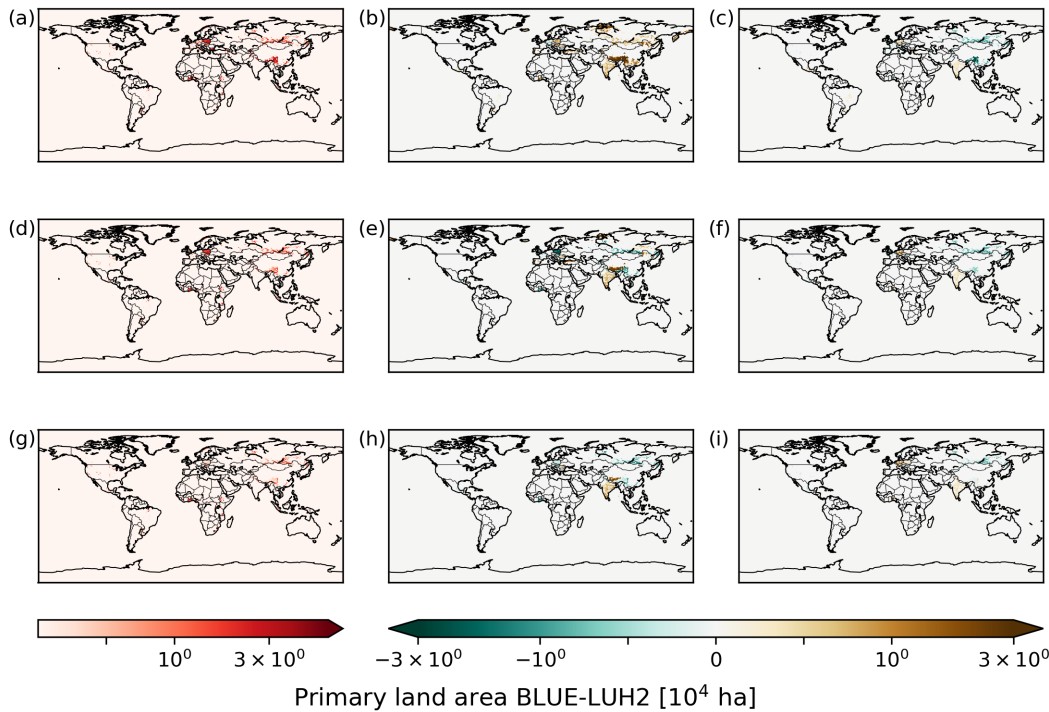

**Figure A5.** Differences in primary land area in BLUE and LUH2 in 2014 for REG850 (a), REG1700 (d), REG1850 (g). Differences HI-REG (b, e, h) and LO-REG (c, f, i) of BLUE-LUH2 primary land area for the same years as in the first column. The global area of primary land in 2014 is $3.6 \cdot 10^9$ ha.



**675** **Appendix B:  Supplementary material**





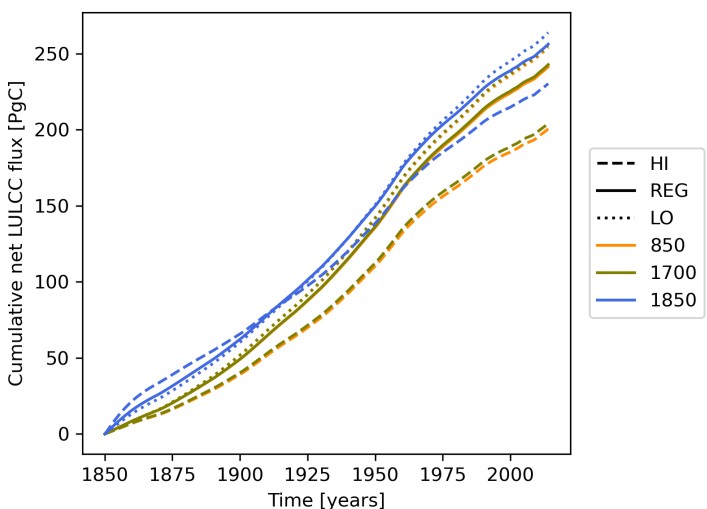

**Figure B1.** As Fig. 2 but with cumulative net LULCC flux over the period 1850–2014.



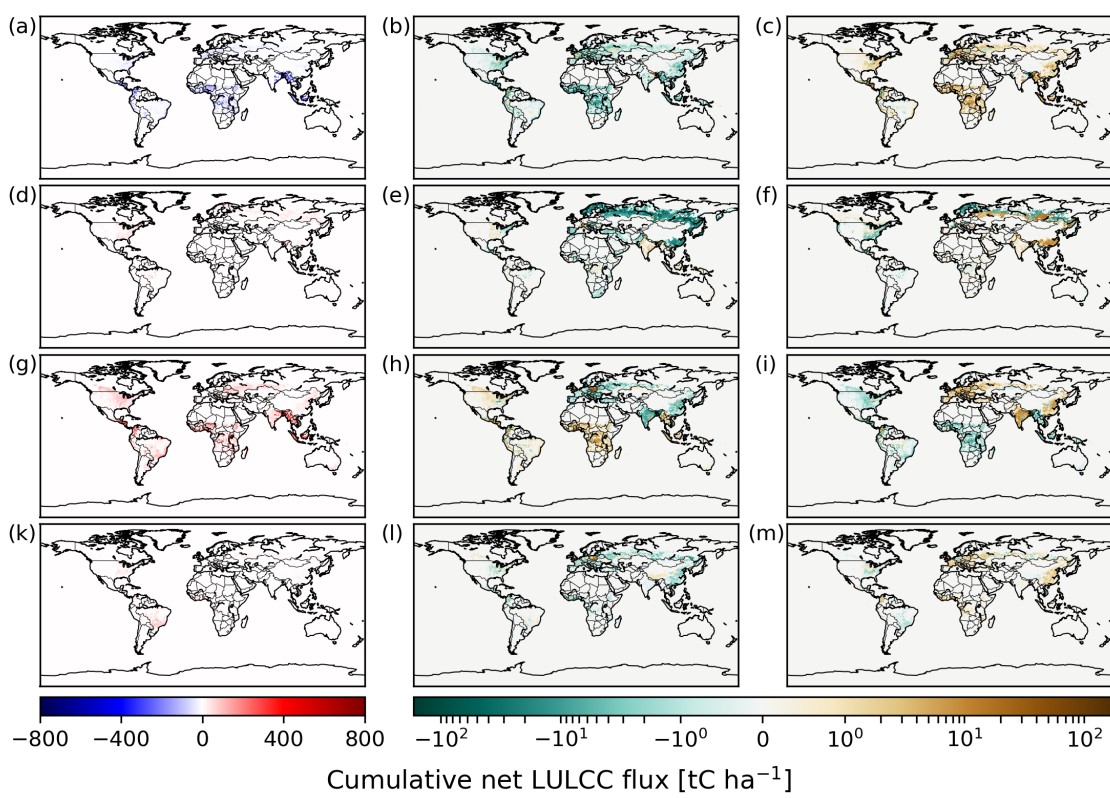

**Figure B2.** As Fig. 5 but for four LULCC activities abandonment (a-c), harvest (d-f), crop (g-i) and pasture (k-m) in the four rows. LULCC scenarios are aggregated in columns: REG1700 in the first column (a, d, g, k), HI1700-REG1700 in the second column (b, e, h, l) and LO1700-REG1700 in the third column (c, f, i, m).





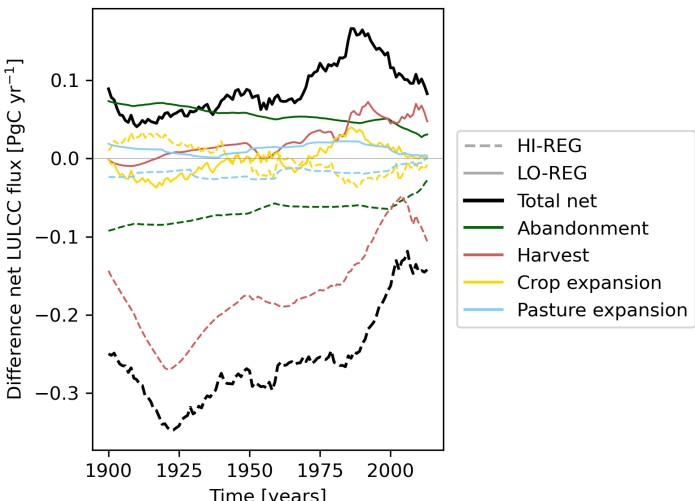

**Figure B3.** Timeseries of differences of the global annual net LULCC flux for the period 1900–2014 from historical simulations. Shown are the differences HI1700-REG1700 and LO1700-REG1700 for the total net LULCC flux and the contribution from each of the four LULCC activities.

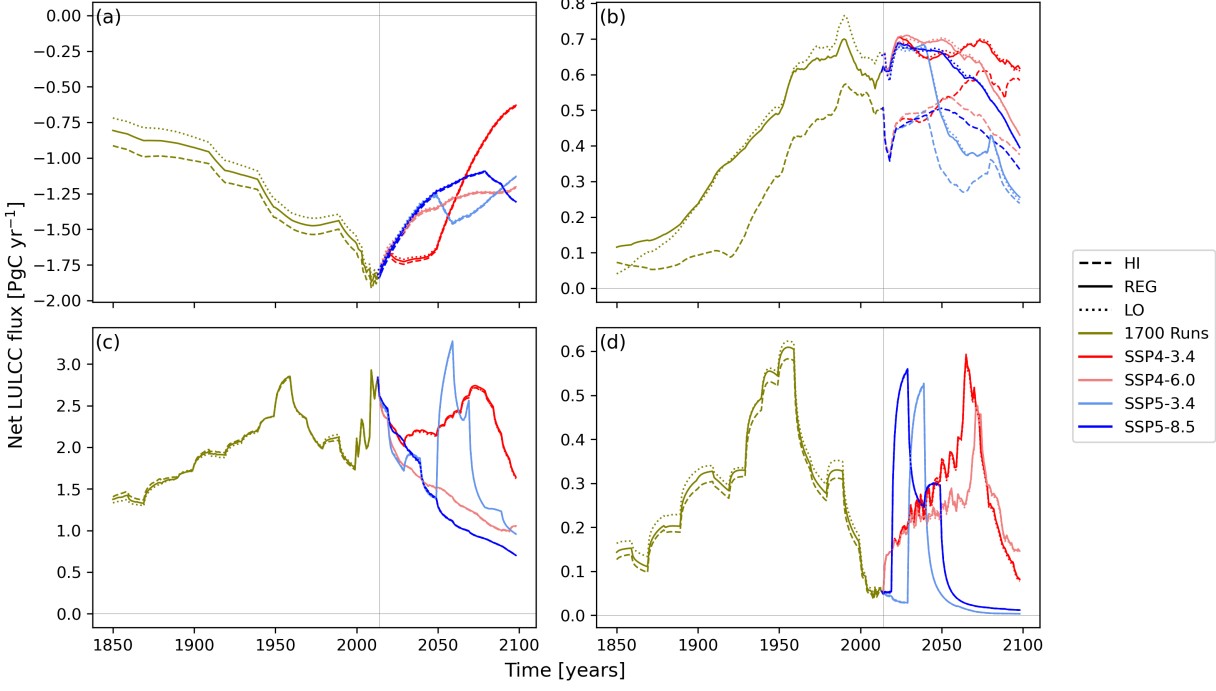

**Figure B4.** As Fig. 1 but showing global annual net emissions for the four LULCC activities (a) abandonment, (b) harvest, (c) crop expansion and (d) pasture expansion. Note the different *y*-axis ranges.