# Peer review of "Bookkeeping estimates of the net land-use change flux – a sensitivity study with the CMIP6 land-use dataset"

_Earth System Dynamics, 2020_

## Author Comment (AC1)

**Net land-use change carbon flux estimates and sensitivities – An assessment with a bookkeeping model based on CMIP6 forcing**

**RC1**

The authors did a set of sensitivity tests using the CMIP6 LULCC forcing (i.e. LUH2) and a bookkeeping model (BLUE) and evaluated the relative importance of different aspects on the uncertainties of LULCC fluxes. It is a very comprehensive analysis with careful simulation design, precise description of different LULCC terms and results. Although the results are only from one model, I believe it documents the details in LULCC carbon flux simulations and answered some common concerns on the uncertainties like the impacts of LULCC, simulation starting time, shifting cultivation and wood harvest. It also gives an important implication that historical LULCC uncertainty is negligible for the LULCC fluxes in the future scenarios. I read the manuscript very carefully and didn't find any major problem. I therefore recommend this work for publication with small modifications.

**Thank you for your feedback. Please find detailed responses to your comments below.**

Some small concerns:

1. The title "...based on CMIP6 forcing" is too broad. The work used only the LULCC forcing (i.e. LUH2). May rephrase and make it more concentrating.

The original title "Net land-use change carbon flux estimates and sensitivities – An assessment with a bookkeeping model based on CMIP6 forcing." was chosen because the use of a bookkeeping model entails that only the LULCC forcing aspect is applied. However, to avoid any confusion we can make the title more specific by writing: "Bookkeeping estimates of the net land-use change flux – a sensitivity study with the CMIP6 land-use dataset".

2. If the authors still want to emphasize "CMIP6" and take this study as a reference analysis to compare with ESMs. It will be helpful if they can provide a short summary of how ESMs considered such aspects in the LULCC estimates (i.e. LULCC, simulation starting time, shifting cultivation and wood harvest).

Thank you for this comment. Most of the reference studies presented here (e.g. Stocker et al., 2011; Wilkenskjeld et al., 2014; Arneth et al., 2017) are actually based on DGVMs and not on bookkeeping models. This will be clarified e.g. in the introduction (lines 42-47) and in Section 3 (paragraph from line 435) to strengthen the link of our results to DGVMs.

ESM CMIP6 simulations coupling DGVMs to other submodels, and e.g. TRENDY simulations with DGVMs as conducted for the global carbon budget (GCB), are based on the same LULCC datasets as used for the bookkeeping model BLUE in this study. Thus the importance of neglecting different parts of the forcing is common to both. However, net land-use change flux estimates might differ due to altered carbon stocks from additional climate feedbacks, both in uncoupled DGVM and coupled ESM simulations. This is already mentioned in the conclusions (from line 467) but can be highlighted even more.

The starting time of a DGVM/ESM/bookkeeping simulation is a typical uncertainty across models. Despite the LUH2 land-use change data being available from 850, CMIP6 simulations start by default in 1850 (Eyring et al., 2016), contributions to the LUMIP intercomparison assess different starting dates of 1700 and 1850 (Lawrence et al., 2016), and 1700 is the common starting year in the more recent TRENDY studies (Friedlingstein et al. 2019, 2020). We will add a comment on this in the manuscript when motivating our corresponding sensitivity experiment.

3. L313-314: "Uncertainty ..." I read this sentence several times but still didn't understand.

The sentence itself ("Uncertainty of wood harvest also explains why REG850 and LO850 produce the same amount of harvest emissions until 1700 (Fig. 2), although their harvest transitions show different extent.") does not contain the relevant information for an explanation. The next sentence "Harvest on primary land is mostly different from REG in the HI setup (Fig. A2), in line with similar net LULCC flux estimates in REG and LO." should be viewed together with the first.

In response to your comment we restructured the text and added more detail: "As mentioned in Section 3.1.1, harvest on primary land mainly results in net fluxes associated with the primary-tosecondary land transitions. The difference in these fluxes when comparing to HI vs. REG setups is much greater than the differences between the REG vs. LO setups (Fig. A2). Similarities in REG and LO harvest on primary land are thus in line with similar net LULCC flux estimates in those experiments. This also explains why REG850 and LO850 produce similar amounts of harvest emissions until 1700 (Fig. 2), although their total harvested area is different."

- 4. L409-410: Did BLUE take bioenergy crops as regular food crops? To me, bioenergy crops are quite different from food crops. What are the possible consequences of taking both as the same? Indeed, bioenergy crops are different in many ways but a unified handling (i.e. crops and bioenergy crops considered as one group) is common to models. While the LUH2 dataset does provide variables for the fraction of crops used for bio-energy purposes, these variables are not used in the preprocessing for BLUE input, and we simply use crop fractions including all crops. As we are not modelling crop harvest but only emissions from land-use change, for example going from crop to secondary land, the differences are likely relatively small. In addition, the fraction of bioenergy crops is small in historical simulations, which we focus our analysis on, and mainly relevant for future scenarios (Hurtt et al., 2020).
- 5. My last point is kind of to echo the merits of this study that I mentioned earlier. The manuscript is written rigorously with a lot of details and supported materials, but it may be too technical for readers who are not very familiar with the bookkeeping models and LULCC carbon flux estimation. I noticed the authors tried to balance it by adding a summary paragraph at the end of each section which is very thoughtful. There still might be some room to improve by e.g. moving some detailed description of figures to supporting information.

Thank you for this suggestion. Details and content that potentially could be moved to the Appendix (for example "Appendix B: Discussion of further features in the results") are

- a) the discussion of crossing points in Figures 3 and 4 (e.g. lines 281-283, 318/319 and 344-346, as "B.1 Discussion of crossing points of net LULCC flux simulations")
- b) Figure 3b and its discussion, i.e. the comparison of sensitivity experiments over their individual simulation lengths (as "B.2 Common reference period of full simulation analysis")
- c) Section 3.1.4 ("B.3 Regional variations of uncertainty")
- d) and the temporal evolution of the experiments 2014-2099 (lines 386-399, as "B.4 Temporal variability in future experiments").

We suggest to separate B.1 and B.2 into a second Appendix to shorten the two longer and relatively complex subsections 3.1.2 and 3.1.3. The sections/paragraphs on spatial patterns and future variability will remain in the main text as they are shorter and directly relevant to our study.

**References:**

Eyring et al.: Overview of the Coupled Model Intercomparison Project Phase 6 (CMIP6) experimental design and organization, Geosci. Model Dev., 9, 1937–1958, https://doi.org/10.5194/gmd-9-1937-2016, 2016.

Friedlingstein et al.: Global Carbon Budget 2019, Earth Syst. Sci. Data, 11, 1783–1838, https://doi.org/10.5194/essd-11-1783-2019, 2019.

Friedlingstein et al.: Global Carbon Budget 2020, Earth Syst. Sci. Data, 12, 3269–3340, https://doi.org/10.5194/essd-12-3269-2020, 2020.

Hurtt et al.: Harmonization of global land use change and management for the period 850–2100 (LUH2) for CMIP6, Geosci. Model Dev., 13, 5425–5464, https://doi.org/10.5194/gmd-13-5425-2020, 2020.

Lawrence et al.: The Land Use Model Intercomparison Project (LUMIP) contribution to CMIP6: rationale and experimental design, Geosci. Model Dev., 9, 2973–2998, https://doi.org/10.5194/gmd-9-2973-2016, 2016.

**RC2**

This manuscript describes a study looking at the contribution of several sensitivities underlying the net LULCC flux by assessing their relative importance using a bookkeeping model (BLUE) based on a LULCC dataset (i.e., LUH2). They compared the impacts of LULCC uncertainty, the starting time of the bookkeeping model simulation, net area transitions versus gross area transitions and neglecting wood harvest on estimates of the net LULCC flux. They also revealed how historical LULCC uncertainty affect the net LULCC fluxes in the future scenarios. This study is very interesting and the results could provide some insight into the sensitivities of net LULCC. Basically, in my opinion some minor issues should be addressed and improved before the manuscript can be published.

**Thank you for your feedback. Please find detailed responses to your comments below.**

1. The article title "Net land-use change carbon flux estimates and sensitivities...based on CMIP6 forcing" is a little misleading. The CMIP6 forcing refers to the LUH2 dataset in the manuscript, so I suggest changing the title to make it more accurate.

Please see reply RC1, comment 1.

**2.** I suggest that Table 2 and 3 can be combined into one table to make it easier to understand the setup of each sensitivity experiment.

This is a helpful suggestion. We plan to separate the tables into one for historical sensitivity experiments (net/NoH/IC/Trans from Table 2 + Table 3) and one for future experiments (the remaining rows from Table 2). This division follows the flow of the paper (Table 1 used in all sections, Table 2 mainly 3.2.2, 3.2.3, 3.2.1, and Table 3 mainly 3.2.2). By reducing the horizontal extent of the columns in Table 3, it should be able to merge it with Table 2.

**3**. Some expressions in the manuscript are difficult to understand, so I suggest the author read through the whole text and rephrase some sentences. For example, L184, "It should be noted that even for the nine main experiments differences between the area evolution in BLUE compared to the LUH2 dataset occur."

We provide a clarification to the sentence in question (a) and also give two further expressions with their possible improvement below:

 a) It should be noted that even for the nine main experiments differences between the area evolution in BLUE compared to the LUH2 dataset occur.
---- Clarification:

It should be noted that the extent of the LULCC areas in BLUE sometimes differs from

the LUH2 input dataset, even for the nine main experiments, mainly because of a mismatch in PFTs between the LUH2 (harvest) input and the BLUE model.

b) Ranges of variability are not equally distributed around the reference simulations. (lines 267ff)

---- Clarification:

The magnitude of the net LULCC flux of HI-REG is often not the same as the REG-LO difference even though the variability of LULCC is asymmetric around REG.

c) Almost no sensitivity of the net LULCC flux to the starting year of the model simulations remains and the relative relationships between the nine main simulations are similar as discussed for cumulative net LULCC flux estimates (Fig. 3). (lines 375 ff).
---- Clarification:

Almost no sensitivity of the net LULCC flux to the starting year of the model simulations remains. The impact of StYr and LULCC uncertainty on the net LULCC flux in 2014 is similar to the characteristics discussed for the cumulative net LULCC flux estimates (Fig. 3).

4. L256-258, panel a-> Fig. 3a; panel b-> Fig. 3b

**Thanks for this comment. We will update the text.**

 L313-314, "Uncertainty of wood harvest also explains why REG850 and LO850 produce the same amount of harvest emissions until 1700 (Fig. 2)". I cannot get the information from the Fig. 2, please confirm if this is about harvest emissions.

**Please see reply to RC1, comment 3.**

6. In the introduction section, the author introduced example studies of Hurtt et al. (2011) and Gasser et al. (2020) in detail, which is good, but I suggest the author add some words about the shortcomings of previous research and the improvements of this work in comparison with the previous studies.

Thank you for your comment. Our work expands on the previous works with bookkeeping models mentioned in the manuscript by using one LULCC dataset providing uncertainty estimates and by performing sensitivity experiments on the link between uncertainties in LULCC (either of initial land cover distribution or of subsequent transitions) and sensitivities in the net LULCC flux. Additionally, we conduct sensitivity experiments to test the relevance of common simulation choices taken in DGVM studies (e.g. in/exclusion of wood harvest or shifting cultivation) in one bookkeeping model. The paragraph beginning with line 68 outlines the ways in which our study builds upon those previous studies, by using LUH2 vs. LUH1 data, using a state-of-the-art bookkeeping model, and incorporating the sensitivity experiments mentioned above. We will update the Introduction with this information to make it clearer.

**RC3**

The manuscript presents a comprehensive analysis of common factors (wood harvest and shifting cultivation, uncertainties of the LULCC dataset, or the initialisation year of the simulation) impacting the LULCC flux estimates of the BLUE bookkeeping model, using factorial experiments to assess their relative importance. It is well written and a very useful contribution. After reading two previous referee comments, I do not have anything else to add, and I recommend acceptance of the manuscript for publication after addressing those minor comments .

Thank you for your feedback. We have responded to the comments by the other two reviewers. Please find our replies in the respective author comments to RC1 and RC2.